# Comparative Measurements and Analysis of the Electrical Properties of Nanocomposites Ti*_x_*Zr_1−*x*_C+α-Cy (0.0 ≤ *x* ≤ 1.0)

**DOI:** 10.3390/ma15227908

**Published:** 2022-11-09

**Authors:** Paweł Żukowski, Piotr Gałaszkiewicz, Vitali Bondariev, Paweł Okal, Alexander Pogrebnjak, Anatolyi Kupchishin, Anatolyi Ruban, Maksym Pogorielov, Tomasz N. Kołtunowicz

**Affiliations:** 1Department of Electrical Devices and High Voltage Technology, Lublin University of Technology, 38A, Nadbystrzycka Str., 20-618 Lublin, Poland; 2Department of Nanoelectronics and Surface Modification, Sumy State University, 2, R-Korsakov Str., 40007 Sumy, Ukraine; 3Physico-Technological Center, Abai Kazakh National Pedagogical University, 13, Dostyk Ave., Almaty 050010, Kazakhstan; 4Medical Institute, Sumy State University, 31, Sanatornaya Str., 40018 Sumy, Ukraine; 5Laboratory of Optical Biosensors and Functional Nanomaterials, Institute of Atomic Physics and Spectroscopy, University of Latvia, 19, Raina Blvd., LV 1586 Riga, Latvia

**Keywords:** nanocomposite, carbides, conductivity, permittivity, tuneling, frequency

## Abstract

In this paper, the frequency-temperature dependence of the conductivity and dielectric permittivity of nc-Ti*_x_*Zr_1−_*_x_*C+α-C*_y_* (0.0 ≤ *x* ≤ 1.0) nanocomposites produced by dual-source magnetron sputtering was determined. The films produced are biphasic layers with an excess of amorphous carbon relative to the stoichiometric composition of Ti*_x_*Zr_1−*x*_C. The matrix was amorphous carbon, and the dispersed phase was carbide nanoparticles. AC measurements were performed in the frequency range of 50 Hz–5 MHz at temperatures from 20 K to 373 K. It was found that both conductivity and permittivity relationships are determined by three tunneling mechanisms, differing in relaxation times. The maxima in the low- and high-frequency regions decrease with increasing temperature. The maximum in the mid-frequency region increases with increasing temperature. The low-frequency maximum is due to electron tunneling between the carbon films on the surface of the carbide nanoshells. The mid-frequency maximum is due to electron transitions between the nano size grains. The high-frequency maximum is associated with tunneling between the nano-grains and the carbon shells. It has been established that dipole relaxation occurs in the nanocomposites according to the Cole-Cole mechanism. The increase in static dielectric permittivity with increasing measurement temperature is indicative of a step polarisation mechanism. In the frequency region above 1 MHz, anomalous dispersion—an increase in permittivity with increasing frequency—was observed for all nanocomposite contents.

## 1. Introduction

Transition metal carbides and metals with high melting points are finding increasing use in engineering. This is due to their high mechanical and thermal properties, as well as their good electrical, capacitive and inductive properties [1,2,3,4]. Nowadays, several methods are known for the fabrication of both bulk materials and metal carbide thin films.

For this purpose, methods such as unpressurised [5] or high-pressure pressing [6], with the simultaneous interaction of high-pressure and high-temperature [7,8] or plasma [9] and pulsed discharge [10], are used. Another promising method is the detonation method [11,12,13]. Coatings produced by this method have high adhesion with the substrate material and high density, as well as the possibility to produce coatings from components with high melting points up to 2800 °C. Coatings produced by this method will have a high quality, provided that the materials applied have sufficiently good mechanical parameters [14]. In order to obtain coatings with high mechanical and electrical parameters, the composition and structure must be selected at the micro and even nanometer levels. Even small changes in both the composition and structure can contribute to significant changes in coating parameters.

The above-mentioned methods of producing metal carbides are used to obtain bulk materials and coatings with thicknesses in the order of 10 µm. A number of carbides have the ability to crumble. To eliminate it, metals with high ductility are used. In the paper [15], multilayered coatings, which consist of alternating layers of carbides and metal oxides, were used to eliminate embrittlement. Each layer has a thickness of the order of 1 µm. The coatings were produced by magnetron sputtering.

Magnetron sputtering makes it possible to obtain micrometer-thick coatings deposited on virtually any substrate, such as metals, semiconductors, ceramics and plastics. A promising direction is the production of highly-entropic materials and the formation of nitrides and carbides with unique properties based on them [16,17,18]. Studies have shown that some combinations of two or three components give composite material layers very good mechanical properties, which are not possible to achieve in solid materials [19,20,21].

The formation of single-phase micro-inclusions that are chemically and structurally disordered has been observed in high-entropy alloys [22,23]. This determines exceptional mechanical properties [24,25,26], high hardness [27,28,29], thermal resistance [30,31,32,33] and excellent electrical and thermal properties [34,35,36,37]. In addition, nitride and carbide layers have low chemical activity and low electron exit work [38]. The most recently developed carbides are MAX-phase. MAX-phase layers are characterised by high hardness and electrical conductivity, have low chemical activity and exit work [38,39,40] and successfully replace gold in the manufacture of electrical contacts because they are wear-resistant.

In this paper [15], two-phase nc-Ti*_x_*Zr_1−_*_x_*C+α-C*_y_* nanocomposites in the concentration range (0.0 ≤ *x* ≤ 1.0) were produced using specially selected parameters of two-source magnetron sputtering. The transition metal carbide nanoparticles had average dimensions of 8 nm to 20 nm, depending on the composition [15], and are the dispersed phase. The second phase constituting the matrix was amorphous carbon. This structure of the fabricated nanocomposites clearly demonstrates that these materials are models for the study of percolation phenomena and hopping conductivity, based on electron tunneling quantum phenomena [41]. The tunneling phenomenon, as it is known, occurs when the dimensions of the potential wells from which the electrons tunnel, and the thicknesses of the barriers separating the wells, are nanometres [42,43,44,45]. In many papers, step conductance studies are performed using direct current—see, for example, [46,47,48,49,50]. The use of AC voltage for testing extends, significantly, the range of information about tunneling processes occurring in nanocomposites [51,52,53,54]. Such studies allow the determination of conduction mechanisms, dielectric relaxation processes, permittivity and polarisation mechanisms in nanocomposites. Investigations of the frequency-temperature electrical properties of nc-Ti*_x_*Zr_1−_*_x_*C+α-C*_y_* (0.0 ≤ *x* ≤ 1.0) nanocomposites may contribute to expanding their range of applications in micro and medium-power electrical devices and power electronics.

The aim of this study was to investigate the basic alternating electrical parameters of nc-Ti*_x_*Zr_1−_*_x_*C+α-C*_y_* nanocomposites in the concentration range (0.0 ≤ *x* ≤ 1.0) at temperatures of 20 K-373 K to determine the mechanisms of conductivity and polarisation and to benchmark the structure and electrical parameters and, on this basis, determine the mechanisms of electron tunneling between the individual structural components of the nanocomposites.

## 2. Materials and Methods

### 2.1. Materials

Nc-Ti*_x_*Zr_1−_*_x_*C+α-C*_y_* (0.0 ≤ *x* ≤ 1.0) nanocomposites were obtained by dual magnetron sputtering [15]. The layers were applied to the planar surface of silicon wafers designed for the manufacture of integrated circuits. The orientation of the wafers was (100) and their thickness was 0.3 mm. First source used five alloy targets with the following compositions: 1—Ti, 2—Ti_0.75_+Zr_0.25_ at. %, 3—Ti_0.5_+Zr_0.5_ at. %, 4—Ti_0.25_+Zr_0.75_ at. %, 5—Zr. The second sputtered target was made of carbon manufactured by Nanoshel (Punjab, India) with a purity of 99.99 %. The operating parameters of both sources were chosen in such a way that the carbon content of the layer was redundant compared to the stoichiometric composition of Ti*_x_*Zr_1−_*_x_*C. The structure and composition of the films produced were investigated by XRD, SIMS and energy-dispersive X-ray spectroscopy [15]. It was determined that the fabricated films are nano-grained nanocomposites with average carbide grain sizes varying from 8 nm to 20 nm, depending on the Ti content. The results of the investigations on the structure and chemical composition of the layers are shown in Table 1. Comparing the composition of the targets and the composition of the fabricated carbides, it can be seen that titanium is more easily atomised than zirconium. From the table, it can be seen that the thicknesses of the produced layers are similar for the different compositions and are approximately 1 µm. Each layer contains surplus carbon. Given the surplus carbon and its density of 2.25 g·cm^−3^ and also that the densities of TiC and ZrC are much higher than those of carbon at 4.91 g·cm^−3^ and 6.73 g·cm^−3^, respectively [55], it should be assumed that the amorphous carbon α-C phase acts as the matrix in the nanocomposites studied. This means that the nanocomposite layers produced are biphasic. In addition, there are further nanometre-sized and even smaller elements on the surface of the nano-grains. These are near-monoatomic carbon films. Their properties differ fundamentally from those of the amorphous carbon matrix [15].

### 2.2. Methods

Determining the electrical properties of nc-Ti*_x_*Zr_1−_*_x_*C+α-C*_y_* nanocomposites has been done using a measuring stand developed at the Department of Electrical Devices and High Voltage Technology at the Faculty of Electrical Engineering and Computer Science at the Lublin University of Technology. The configuration and functional principle of the stand was described in publications [56,57,58]. Measurements were carried out as a function of frequency in the range from 50 Hz to 5 MHz and temperature range from 20 K to 373 K. Time needed for a single sample measurement was approximately 5 h. In order to improve the efficiency of the measuring stand, two impedance meters measure two samples simultaneously. The stand is fully automated. Frequency and temperature settings, as well as saving of measurement results, are made by a dedicated computer programme.

Measurements were carried out in a capacitor configuration, of which the schematic is shown in Figure 1. One plate of the capacitor was a silicon substrate (2), onto which the nanocomposite layer (1) was deposited. A second plate made of silver paste (3) was applied to the top surface of layer (1). A silver paste electrode (4) was also applied to the bottom surface of the silicon wafer. That allowed the elimination of the transition resistance between the measuring contact and the silicon plate. The capacitor configuration of the electrodes allows both conduction and capacitive currents to be measured, and on their basis, the conductivity and permittivity of the layer are to be determined.

Current flow in materials is described by Maxwell’s second equation in differential form [59]:(1)$$\nabla \times \vec{H}={\vec{j}}_{R}+{\vec{j}}_{C}$$
where: $\vec{H}$—magnetic field strength vector, ${\vec{j}}_{R}$—conduction current density, ${\vec{j}}_{C}$—shift current density.

Alternating current measurements use a sinusoidal electric field to force current flow:(2)$$\vec{E}={\vec{E}}_{0}\sin\left(\omega t\right)$$
where: $\vec{E}$—electric field strength vector, ${\vec{E}}_{0}$—electric field amplitude vector, *ω* = 2*πf*—circular frequency, *f*—frequency, *t*—time.

The densities of conduction and shift currents in a sinusoidal electric field—Equation (1), are described by equation [59]:(3)$${\vec{j}}_{R}(\omega )=\sigma (\omega )\vec{E}=\sigma (\omega ){\vec{E}}_{0}\sin(\omega t)$$
where: *σ*(*ω*)—conductivity, *t*—time.
(4)$${\vec{j}}_{C}\left(\omega \right)=\omega \cdot \varepsilon^{\prime }\left(\omega \right)\cdot \varepsilon_{0}\cdot {\vec{E}}_{0}\sin\left(\omega t-\frac{\pi }{2}\right)$$
where: *ε*′(*ω*)—dielectric permittivity of the material, *ε*_0_—dielectric permittivity of a vacuum.

From Equations (1), (3) and (4), it can be concluded that the current flow is determined by two material parameters. Conductivity determines the electrical conductivity, while permittivity characterises the polarisation of the material. Other parameters determined in AC measurements, such as, for example, the tangent of the loss angle tanδ or the imaginary permittivity, are derivatives of these material parameters. Therefore, in this paper, measurements of the basic AC parameters of nanocomposite films—conductivity and real permittivity—were performed in a parallel equivalent scheme. The frequency coefficients of the rate of change of conductivity and the dependence of the imaginary permittivity on the real permittivity were also determined.

## 3. Hopping Conductivity of Nanocomposites Considering Quantum Mechanical Electron Tunneling Phenomenon

To analyse the AC properties of the fabricated nanocomposites, a hopping conductivity mechanism was used, which involves tunneling electrons between potential wells. The work uses a tunneling mechanism between potential wells—the nearest neighbours [41]. Models describing this phenomenon have been developed since the 1950s. Initially, research focused on conducting direct current. Then, models were developed to describe the conduction of alternating current, including, e.g., [42]. According to this model, the frequency dependence of conductivity is described by the equation:(5)$$\sigma \left(f\right)∼f^{S}$$
where: *S* = const ≤ 0.8.

The publication [60] developed a model describing both DC and AC conductivity of nanocomposites based on electron tunneling. Based on this, the conductivity analysis was performed in GaAs semiconductors irradiated with ions. In the article [61] the model was further developed. The works [56,61] take into account that in nanocomposites with conductivity by tunneling, the dielectric relaxation time should be taken into consideration. The occurrence of the relaxation time is based on the fact that an electron, after tunneling from one well of potential to another, does not immediately perform next tunneling. It must remain in the potential well for some time τ, named relaxation time [62]. Taking into account the relaxation time causes the frequency coefficient in the Equation (5) to become a frequency function [62]:(6)$$\sigma \left(f\right)∼f^{\alpha \left(f\right)}$$

Values *α*(*f*) can be determined from the differentiation of *σ*(*f*) with numerical methods using the equation:(7)$$\alpha \left(f\right)=\frac{d\left[\log\sigma \left(f\right)\right]}{d\left(\log f\right)}$$

The model assumes that in nanocomposites the tunneling valence electrons are located in nanometre potential wells at the highest occupied levels. (Figure 2). For tunneling to occur, the valence electron should be thermally excited to the lowest unoccupied state (arrow A in Figure 2). This is because direct tunneling from a staffed state in one well to a staffed state in another well is prohibited by the Pauli exclusion principle [63]. The difference in energy between the highest occupied well and the lowest unoccupied well is the tunneling activation energy (conductivity activation energy).

Because the potential wells are nanometres apart, it is possible for electrons to tunnel between them. Under the influence of an alternating electric field (Figure 2), a current (1) is generated on the tunneling path between the left and centre potential wells, the density of which is:(8)$$j_{1}=\sigma_{0}\cdot E=\sigma_{0}\cdot E_{0}\sin\omega \cdot t$$
where: *E*—the intensity of the external alternating electric field, *E*_0_—electric field strength amplitude, *σ*_0_—conductivity, *ω*—angular frequency, *t*—time.

After tunneling, an electric dipole is formed because the left well, due to the lack of an electron, becomes positively charged. In contrast, the middle well acquires an extra electron and becomes negatively charged. This causes additional polarisation of the nanocomposite [64]. After tunneling to the central well, the electron remains there for the relaxation time *τ*. After this time, two scenarios for the next tunneling are possible. In the first one, the electron tunnels from the middle to the right well with probability *p*. This gives a second component of the current density (2), phase shifted in relation to an alternating electric field—Figure 2:(9)$$j_{2}=\sigma_{0}\cdot E_{0}\cdot p\sin\left[\omega (t-\tau )\right]$$

In the second case, the electron tunnels from the middle to the left well with probability (1 − *p*). This causes the appearance of the third component of the current density (3), also shifted in phase—Figure 2:(10)$$j_{3}=-\sigma_{0}\cdot E_{0}(1-p)\sin\left[\omega (t-\tau )\right]$$

From Equations (8)–(10), it follows that the real component of the tunneling current density is:(11)$$j_{r}=\sigma_{0}E_{0}\left[1-(1-2p)\cos\left(\omega t\right)\right]\sin\left(\omega t\right)$$

The DC and low-frequency conductivities for tunneling are:(12)$$\sigma_{DC}=\sigma_{L}=2p\sigma_{0}$$

In the high-frequency area:(13)$$\sigma_{H}=\sigma_{0}$$

In contrast, there is a frequency dependence of the current density in the transition area, described by Equation (6). The value of the relaxation time is not a constant value and depends on the distance between the potential wells [62].

Due to the random distribution of the potential wells in the nanocomposite, there is a probability distribution of relaxation times that should be determined only for positive values. Negative values would mean that the electron returned from the middle to the left well even before that tunneled from the left well. This corresponds to the Landau distribution in the Moyal approximation [65]:(14)$$F_{LM}(\tau )=\frac{1}{\sigma_{m}\sqrt{2\pi }}\exp\left(-\frac{\tau -\tau_{m}}{2\sigma_{m}}-\frac{1}{2}\exp\left(-\frac{\tau -\tau_{m}}{\sigma_{m}}\right)\right)$$
where: *σ_m_*—standard deviation; *τ_m_*—expected value.

The Equation (11) for the density of the real AC component takes the form:(15)$$j_{r}\left(\omega \right)=\sigma_{0}E_{0}\left[1-(1-2p)\right]\int_{\tau }F_{LM}\left(\tau \right)\cos\omega \tau d\tau$$

Numerical methods were used to determine the real current density component *j_r_* in accordance with Equation (15). The calculations were performed for probability values *p* ranging from 10^−6^ to 0.5 with a small step. Figure 3a presents an example of a relationship *j_r_* (*f · τ_m_*).

It can be seen from Figure 3a that in the low-frequency area, the conductivity does not depend on frequency, and its value is equal to the constant-current value—Equation (12). The frequency causes an increase in conductivity, described by Equation (6). In the high-frequency area, the conductivity also becomes constant. Examples of dependencies of the *α*(*f*), coefficient values entered into Equation (6), were obtained by numerical differentiation in accordance with Equation (7) and shown in Figure 3b. The figure shows that as the jump probability *p* decreases, the maximum value of the frequency factor α*_max_* increases, and its position on the *x* axis shifts to the lower frequencies area. The parameters that fully characterise the frequency dependence of conductivity for electron tunneling are the jump probability *p*, the expectation value of the relaxation time *τ_m_* o and the value of the conductivity in the high frequency area σ_0_. From these, using Equations (15) and (7), the waveforms *σ*(*f*) and *α*(*f*) can be calculated. There is more than one tunneling mechanism in the nanocomposite, which differs in *p*, *τ_m_* and *σ*_0_ values, and a corresponding number of maxima appear on the *α*(*f*) relationship [15]. Experimental verification of the model was been carried out in many studies; see, e.g., [66,67].

## 4. Results and Discussion

### 4.1. Frequency-Temperature Dependence of the Conductivity of nc-Ti_x_Zr_1−x_C+α-C_y_ Nanocomposites

Figure 4 shows the frequency dependence of the conductivity and frequency coefficients for the nc-TiC+α-C*_y_* nanocomposite for selected measurement temperatures. We will now compare the experimental results, shown in Figure 4, with the results of the electron tunneling conductivity model—Figure 3. From the figures, it can be seen that in the low-frequency area, the conductivity hardly depends on the frequency. It is a direct current conductivity. An increase in temperature causes an increase in conductivity. This means that there is a dielectric-type conductivity in the nanocomposite. As the frequency increases above 10^3^ Hz, the conductivity starts to increase. Conductivity waveforms with increasing temperature shift to the area of higher frequencies. This is related to the reduction of relaxation times with increasing temperature. The *σ*(*f*) waveforms show slight inflection in the frequency area of about 10^5^ Hz.

In order to visualise the inflections, numerical differentiation according to the Equation (7) was used. Figure 4b shows the dependence of the frequency coefficient α(*f*) for the nc-TiC+α-C*_y_* nanocomposite. The figure shows that in the frequency range above 10^5^ Hz there are two maxima very close to each other. In the low frequency area, around 200 Hz, at low temperatures, there is a third, much weaker maximum. The similarity of the waveforms of both conductivity and frequency factor, shown in Figure 3 and Figure 4, implies that the observed relationships are consistent with the hopping mechanism of charge transfer, considering the quantum phenomenon of electron tunneling described in Section 3. This indicates the presence of a hopping conduction mechanism in the nanocomposite. The occurrence of three maxima means that there are at least three types of tunneling in the material, differing in the values of the relaxation times.

Figure 5 shows the frequency dependence of the conductivity σ(*f*) and the frequency factor α(*f*) for the nc-Ti_0.86_Zr_0.14_C+α-C*_y_* nanocomposite. The introduction of zirconium into the nanocomposite resulted in bends becoming apparent in the conductivity waveforms (Figure 5a). The frequency dependence of the α(*f*) coefficient (Figure 5b) shows two distinct maxima in the investigated frequency range. Their position and maximum values depend on the measurement temperature. The value of the maximum in the mid-frequency area increases with increasing temperature, and its position shifts to the higher-frequency area. The high-frequency maximum, on the other hand, decreases with increasing temperature and shifts to the higher-frequency area much less than the low-frequency maximum.

After increasing the zirconium content to 0.25 (Figure 6), three maxima are observed. These maxima behave with increasing frequency and temperature, similar to higher titanium contents. The third maximum is the most marked for this zirconium content. It occurs in the area of low frequencies, about (300–800) Hz at temperatures below 200 K. A careful analysis of Figure 4, Figure 5, Figure 6, Figure 7 and Figure 8 shows that this maximum, although less pronounced, also occurs in the remaining nanocomposites tested.

Two successive compositions of nanocomposites with zirconium contents of 0.61 (Figure 3) and 1.0 (Figure 4) show similar frequency-temperature dependencies of the conductivity *σ*(*f*) and the frequency coefficient *α*(*f*).

For all titanium contents, an increase in low-frequency conductivity is observed with increasing temperature. This means that nc-Ti*_x_*Zr_1−_*_x_*C+α-C*_y_* nanocomposites in the entire range of titanium concentration changes of 0.0 ≤ *x* ≤ 1.0 exhibit dielectric-type conductivity.

It can be seen in Figure 4, Figure 5, Figure 6, Figure 7 and Figure 8 that there are three maxima on the dependencies α(*f*) for all titanium contents in the tested nc-Ti*_x_*Zr_1−*x*_C+α-C*_y_* nanocomposites. This means that in the investigated frequency range, three tunneling mechanisms are observed, differing in the values of relaxation times and the effect of temperature on them. Low and high frequency peaks decrease in value with increasing temperature. This is due to an increase in the electron tunneling probability *p* from the middle well to the right well (compare Figure 3 and Figure 4, Figure 5, Figure 6, Figure 7 and Figure 8). On the other hand, the increase in the maximum value of the coefficient α(*f*) for the maximum in the medium frequencies area is caused by the reduction of the probability of tunneling *p* with increasing temperature.

In order for the nanocomposite to have three tunneling mechanisms, there should be two types of wells of potential. We will label these wells as P_1_ and P_2_. Then, the following tunneling mechanisms will be possible: P_1_ ↔ P_1_, P_2_ ↔ P_2_, P_1_ ↔ P_2_. One type of well is metal carbide nanograin. It was established in [15] that on the surface of nanograins there is a transition layer consisting of carbon with a structure significantly different from that of the amorphous carbon matrix. Since the matrix is amorphous carbon, it can be assumed that the second type of potential well is the carbon coatings on the surface of carbide nanograins. Due to the fact that the volume of such coatings is smaller than that of metal carbide grains, the coatings are probably responsible for a weak peak in the low frequency area (P_2_ ↔ P_2_). Comparing the temperature dependence of the values of the maxima, it should be assumed that the maximum in the area of medium frequencies is caused by tunneling between the nanograins (P_1_ ↔ P_1_). The transitions between nanograins and coatings (P_1_ ↔ P_2_) are responsible for the high-frequency maximum.

### 4.2. Frequency-Temperature Dependencies of the Permittivity of Ti_x_Zr_1−x_C+α-C_y_ Nanocomposites

In this paper, the frequency-temperature dependencies of the dielectric permittivity were determined. It turned out that these relationships for different nanocomposite compositions are very similar. Therefore, as an example, Figure 9a and Figure 10a show the permittivity dependencies for nanocomposites with low and high zirconium content (nc-Ti_0.86_Zr_0.14_C+α-C*_y_* and nc-Ti_0.39_Zr_0.61_C+α-C*_y_*).

As can be seen from the frequency dependence in the low-frequency and high-temperature range, permittivity hardly depends on frequency. As frequency increases, there are two stages of permittivity reduction. The low-frequency stage moves rapidly into the higher-frequency region as the temperature increases. The high-frequency stage moves much more slowly. This means that the stages differ not only in the values of the relaxation times but also in the influence of the temperature values on them. In the low-frequency and higher-temperature regions, the permittivity value hardly depends on the frequency. This is known as static permittivity. There is an inflection in the mid-frequency region, caused by the transition from one polarisation mechanism to another. As can be seen from Figure 9 and Figure 10, the inflection between the low- and high-frequency stages is most noticeable at 319 K. Therefore, we will analyse, as an example, the waveform for 319 K. In the low-frequency region, the permittivity is the sum of the permittivity of the static low-frequency and high-frequency stages. With increasing frequency, there is a reduction in the permittivity of the low-frequency stage. This ends in inflection at frequencies of around 4 × 10^4^ Hz.

The value of the permittivity at this frequency is referred to as the static permittivity for the high-frequency stage. This means that the value of the static permittivity for the low-frequency stage is about 30. The value of the static permittivity for the high-frequency stage is also about 30. As is known, the expectation value of the dielectric relaxation time is related to the frequency at which the permittivity decreases twice from the value of *ε_S_* (static permittivity) to the value of *ε_S_*/2 [68]:(16)$$\tau =\frac{1}{2\pi f_{1/2}}$$
where: *τ*—relaxation time, *f*_1/2_—frequency at which permittivity decreases twice.

From Figure 9, the frequencies at which the permittivity values decrease two times at 319 K are determined. For the low-frequency stage, this is about 2 × 10^4^ Hz, and for the high-frequency stage about 4 × 10^5^ Hz. Based on Equation (16) the expected values of the relaxation times were calculated to be about 8 × 10^−6^ s and about 4 × 10^−7^ s, respectively. As the temperature decreases below 319 K, the relaxation times increase, while above the temperature of 319 K they decrease. Similar values of relaxation times and their temperature variation are found for the nanocomposite with higher zirconium content. It can be seen from Figure 9a and Figure 10a that the static permittivity values for both stages increase with increasing temperature. The observed increase in static dielectric permittivity due to increasing temperature is indicative of a hopping polarisation mechanism caused by tunneling. In this mechanism, the concentration of dipoles formed by tunneling increases with increasing temperature [64]. When solid dipoles are present in the materials (orientation polarisation), the permittivity is described by the Debye formula [68], and the value of the static permittivity decreases with increasing temperature as 1/*T*.

In order to clarify the dielectric relaxation mechanisms occurring in the studied nanocomposites, the dependence of the imaginary component of the permittivity on the real component, the so-called Cole-Cole diagrams, was determined. As is known, the imaginary component of the complex permittivity is calculated from the equation [68]:(17)$$\varepsilon^{\prime \prime }\left(f\right)=\frac{\sigma \left(f\right)}{\omega \varepsilon_{0}}$$
where: *σ*(*f*)—conductivity, *ω* = 2*πf*—circular frequency, *ε*_0_—dielectric permittivity of a vacuum.

The values of the imaginary permittivity were calculated from the *σ*(*f*) waveforms (Figure 4, Figure 5, Figure 6, Figure 7 and Figure 8) using Equation (17). The dependence of *ε*″(*ε*′) for selected measurement temperatures is shown in Figure 9b and Figure 10b. In the case of dielectric relaxation, a number of mechanisms are known to describe the *ε*″(*ε*′) dependence. The best known among these is the Debye relaxation mechanism. Half-circle-shaped *ε*″(*ε*′) waveforms are characteristic of this mechanism [68]. The Debye mechanism occurs when the relaxation time *τ*(*f*) = constant. The Cole-Cole relaxation mechanism [69,70] is observed when there is a distribution of relaxation times around the expectation value *τ_m_*. For this mechanism, the graph of *ε*″(*ε*′) assumes an arc shape. The value at the maximum of this relationship is smaller than the width of the arc on the *x*-axis. The Davidson-Cole mechanism is also applicable [71,72]. For this mechanism, an asymmetric *ε*″(*ε*′) dependence graph is characteristic. The asymmetric empirical model of Havriliak and Nagami [73] is also used for dielectric relaxation analysis. It can be seen from Figure 9b and Figure 10b that three symmetrical maxima are evident in the Cole-Cole diagrams over the studied temperature range. The maximum in the area of small values of the permittivity *ε*′ (Figure 10) has been truncated due to the end of the meter range. The values in the maxima are much smaller than their widths. The relationships seen in Figure 9b and Figure 10b are characteristic of the Cole-Cole dielectric relaxation mechanism. This means that there are probability distributions of relaxation times in the nanocomposites. This is consistent with the electron tunneling conductivity model presented in Section 3.

In the frequency region above 1 MHz, anomalous dispersion—an increase in the value of the real permittivity with increasing frequency—was observed for all nanocomposite compositions. In the Cole-Cole diagrams of the dispersion dependence of the anomalous imaginary permittivity on the real permittivity *ε*″(*ε*′)—Figure 9b and Figure 10b look practically like straight lines with a large slope to the *ε*′ axis. Unfortunately, the explanation of the mechanism of the anomalous dispersion is hampered by the truncation of its waveform by the upper range of the measuring meter (5 MHz).

## 5. Conclusions

In this paper, the frequency-temperature dependence of the conductivity and dielectric permittivity of nc-Ti*_x_*Zr_1−_*_x_*C+α-C*_y_* (0.0 ≤ *x* ≤ 1.0) nanocomposites produced by dual-source magnetron sputtering was determined. The samples contained an excess of amorphous carbon phase compared to the stoichiometric composition of Ti*_x_*Zr_1−_*_x_*C. Consequently, the films produced are biphasic layers. The matrix in the nanocomposites studied was amorphous carbon and the dispersed phase is metal carbide nanoparticles. Measurements were made in the frequency range 50 Hz–5 MHz at temperatures from 20 K to 373 K. It was found that the conductivity and permittivity of the nanocomposites are due to electron tunneling. Three maxima are observed on the frequency dependence of the conductivity coefficient. The low-frequency maximum is related to the tunneling of electrons between carbon coatings found on the surface of metal carbide nanoshells. The structure of such coatings differs from that of amorphous carbon. The mid-frequency maximum is due to electron transitions between metal carbide nanoshells. The high-frequency maximum is associated with tunneling between the nano-grains and the carbon coatings.

Cole-Cole dipole relaxation was found to occur in the nanocomposites, characterised by a distribution of relaxation times around the expected value. An increase in static dielectric permittivity was observed with increasing measurement temperature. This is indicative of a step polarisation mechanism, whereby the concentration of dipoles, formed by tunneling, increases with increasing temperature. In the frequency region above 1 MHz, an anomalous dispersion of the true permittivity was observed for all nanocomposite compositions—an increase in value with increasing frequency. Elucidation of the mechanism of anomalous dispersion in the studied nanocomposites is hampered by the fact that its waveform is truncated by the upper range of the measuring meter of 5 MHz.

## Figures and Tables

**Figure 1 materials-15-07908-f001:**
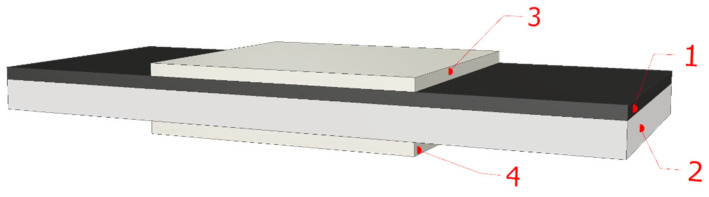
Cross-section of Ti*_x_*Zr_1−_*_x_*C+α-C*_y_* nanocomposite sample: 1—nanocomposite layer, 2—silicon substrate (bottom plate), 3—silver paste top plate, 4—silver paste electrode.

**Figure 2 materials-15-07908-f002:**
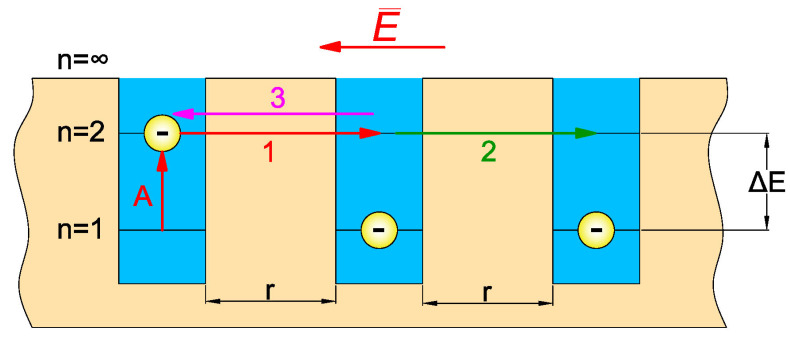
Potential wells: A—excitation of the electron from the highest occupied state to the lowest unoccupied state, 1—tunneling of the electron from the left well to the middle well, 2—tunneling from the middle well to the right well, 3—return tunneling from the middle well to the left well, ∆*E*—distance between occupied and unoccupied states or activation energy of tunnelling, r—distance between wells of potential.

**Figure 3 materials-15-07908-f003:**
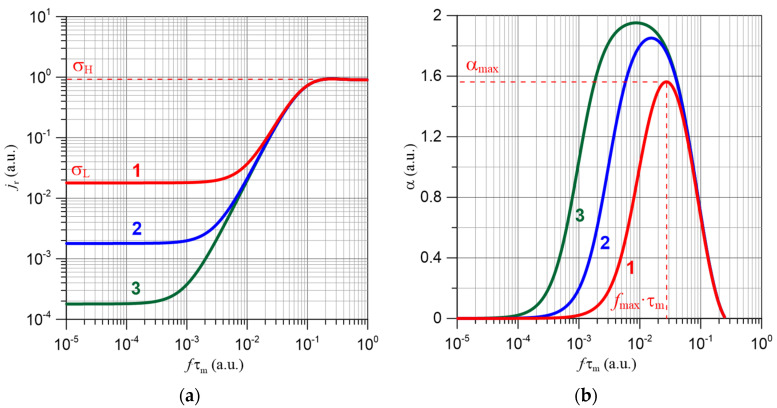
Relationship of the actual component of current density *j_r_*—(**a**) and frequency coefficient α(*f*)—(**b**) as a function of the product of the frequency value *f* and the expected value of the relaxation time *τ_m_*. Computer simulation for: 1—*p* = 0.01; 2—*p* = 0.001; 3—*p* = 0.0001.

**Figure 4 materials-15-07908-f004:**
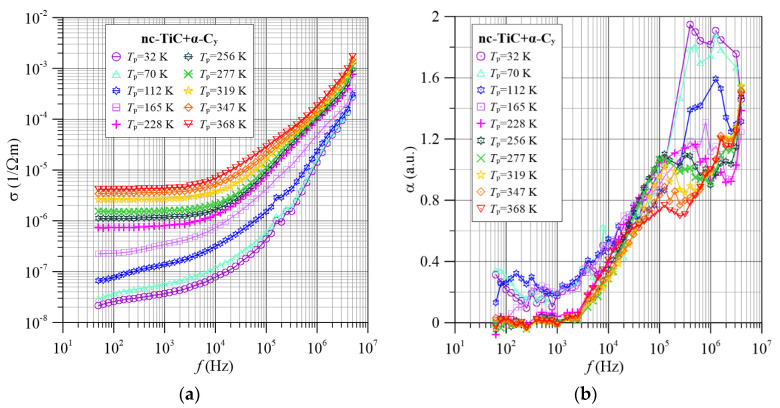
Frequency dependence of conductivity (**a**) and frequency factor α(*f*) (**b**) for selected measurement temperatures for the nc-TiC+α-C*_y_* nanocomposite.

**Figure 5 materials-15-07908-f005:**
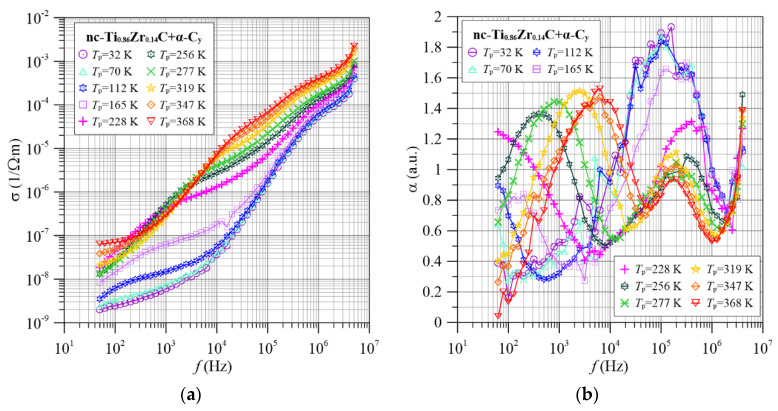
Frequency dependence of conductivity (**a**) and frequency factor α(*f*) (**b**) for selected measurement temperatures for the nc-Ti_0.86_Zr_0.14_C+α-C*_y_* nanocomposite.

**Figure 6 materials-15-07908-f006:**
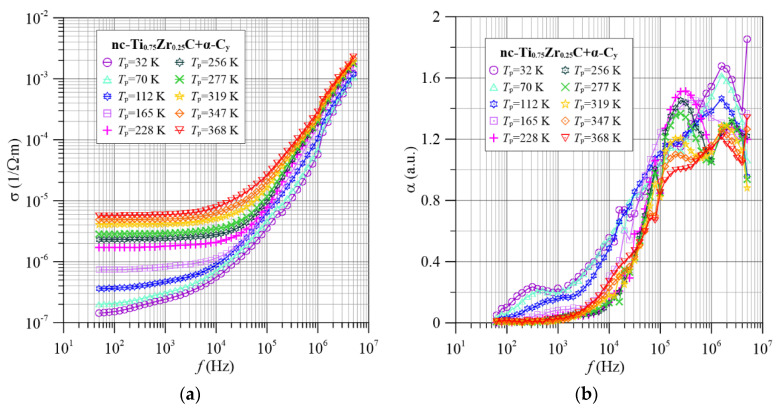
Frequency dependence of conductivity (**a**) and frequency factor α(*f*) (**b**) for selected measurement temperatures for the nc-Ti_0.75_Zr_0.25_C+α-C*_y_* nanocomposite.

**Figure 7 materials-15-07908-f007:**
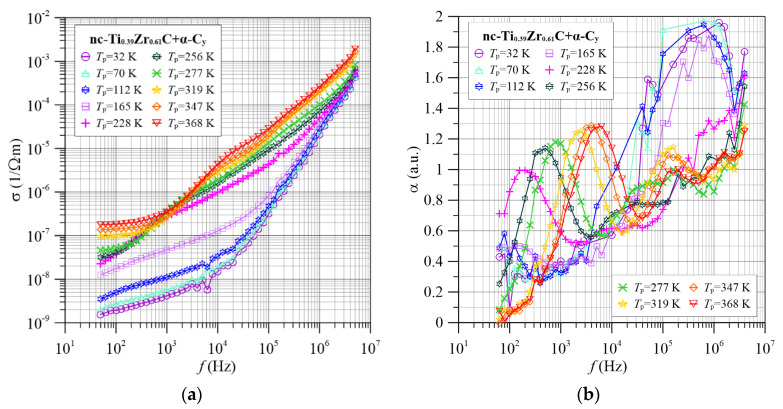
Frequency dependence of conductivity (**a**) and frequency factor α(*f*) (**b**) for selected measurement temperatures for the nc-Ti_0.39_Zr_0.61_C+α-C*_y_* nanocomposite.

**Figure 8 materials-15-07908-f008:**
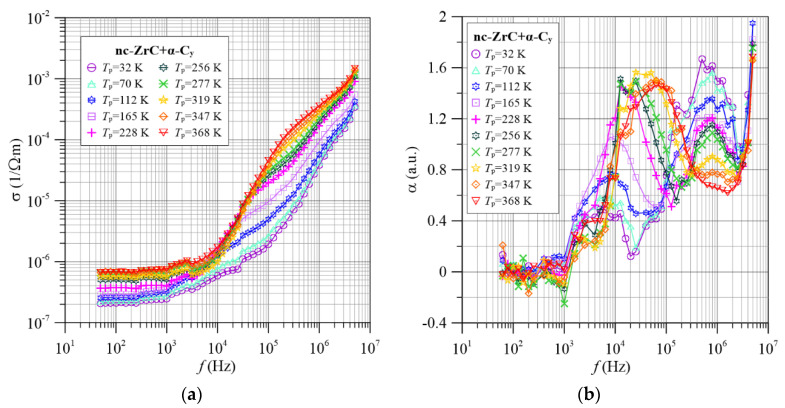
Frequency dependence of conductivity (**a**) and frequency factor α(*f*) (**b**) for selected measurement temperatures for the nc-ZrC+α-C*_y_* nanocomposite.

**Figure 9 materials-15-07908-f009:**
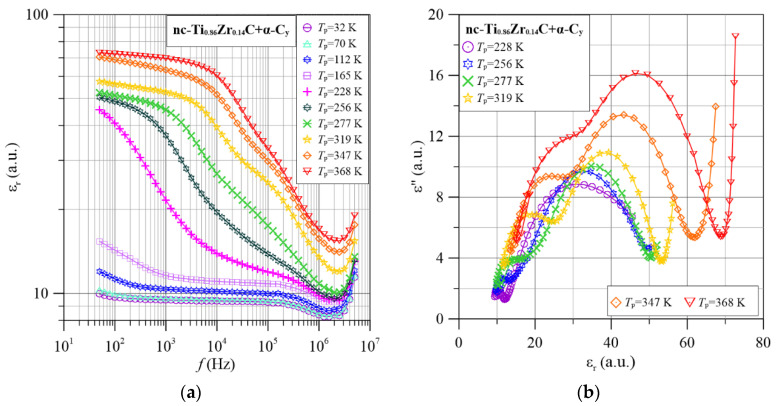
Frequency dependence of permittivity (**a**) and Cole-Cole diagrams (**b**) for selected measurement temperatures for the nc-Ti_0.86_Zr_0.14_C+α-C*_y_* nanocomposite.

**Figure 10 materials-15-07908-f010:**
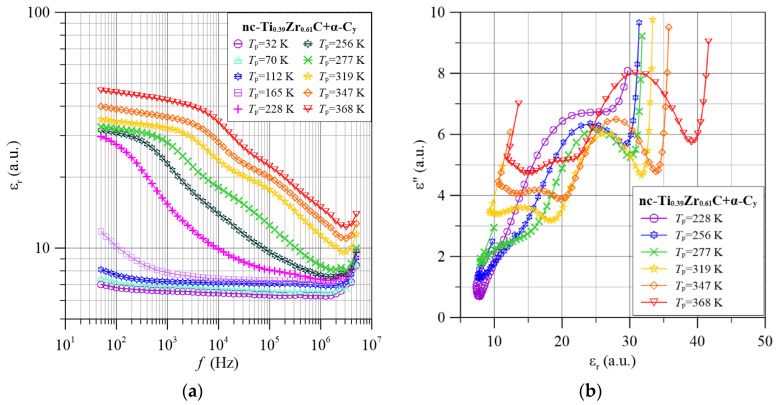
Frequency dependence of permittivity (**a**) and Cole-Cole diagrams (**b**) for selected measurement temperatures for the nc-Ti_0.39_Zr_0.61_C+α-C*_y_* nanocomposite.

**Table 1 materials-15-07908-t001:** Sub-structural parameters of nc-Ti_x_Zr_1−*x*_ C+α-C coatings, based on data from [15].

Sample Number	Target Composition	Real Sample Composition and Structure	Surplus C, a.u.	Thickness, µm	Grain Size *L*, nm	Thickness of the Carbon Coating on the Nanoparticles Surface, nm
1	Ti; C	nc-TiC+α-C_1.25_	1.25	0.984	18 ± 2	0.3–0.5
2	Ti_0.75_+Zr_0.25_; C	nc-Ti_0.86_Zr_0.14_C+α-C_2.62_	2.62	1.165	9 ± 2	
3	Ti_0.5_+Zr_0.5_; C	nc-Ti_0.75_Zr_0.25_C+α-C_0.95_	0.95	1.097	11 ± 2	
4	Ti_0.25_+Zr_0.75_; C	nc-Ti_0.39_Zr_0.61_C+α-C_0.21_	0.21	1.132	15 ± 2	
5	Zr; C	nc-ZrC+α-C_0.20_	0.20	1.015	20 ± 2	

## Data Availability

Not applicable.

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
