# Peer review of "Comparative Measurements and Analysis of the Electrical Properties of Nanocomposites TixZr1−xC+α-Cy (0.0 ≤ x ≤ 1.0)"

_materials, 2022, doi:10.3390/ma15227908_

Round 1

Reviewer 1 Report

Å»ukowski et al. have presented the manuscript titled: Comparative measurements and analysis of the electrical properties of nanocomposites TixZr1-xC+α-Cy (0.0 ≤ x ≤ 1.0). Overall presentation of the work is good, but there are few suggestions which I think are necessary to explain before publication.

1.      In the topic, abstract and fabrication, authors have described the different x values, (abstract 0.39 ≤ x ≤ 1.0, topic 0.0 ≤ x ≤ 1.0), any specific reason?

2.      Abstract is so much general, I suggest the authors to highlight their achieved results (values) in the abstract section to make the article more attractive.

3.      Please highlight the specific reason of performing this study? What were the drawbacks in previous research work in the last portion of introduction?

4.      In the material section please mention the specifications of the XRD analysis, θ step size, scan time, machine model, country etc.

5.      Moreover, XRD analysis is missing in the results section. In the study of materials it is incomplete without the structural, purity and impurity analysis. Please add the result.

6.      I think it’s better to confirm your claim of successful fabrication add the SEM images along with the EDS spectra.

7.      In Figure 4, authors have to explain the reason of such variations in the Frequency-temperature dependence of the conductivity of nc-TixZr1-xC+α-Cy plots. What factor has brought these changes?

8.      Conclusion is so much comprehensive, it should be concise covering all the aspects of performed research.

Author Response

Please see in the attach.

Reviewer 2 Report

I find that the manuscript entitled: "Comparative measurements and analysis of the electrical properties of nanocomposites TixZr1-xC+α-Cy (0.0 ≤ x ≤ 1.0)" is an interesting.

In this paper is presented the frequency-temperature dependence of the conductivity and dielectric permittivity of nc-TixZr1-xC+α-Cy (0.39 ≤ x ≤ 1.0) nanocomposites produced by dual-source magnetron sputtering. The films produced are biphasic layers with an excess of amorphous carbon relative to the stoichiometric composition of TixZr1-xC.

The authors presented the conductivity and permittivity dependencies of the frequency range through three tunneling mechanisms differing in relaxation times. The low-frequency and high-frequency maxima decrease with increasing temperature and their position shifts to the higher frequency region. The conductivity maximum in the mid-frequency region also shifts to higher frequencies and its value increases with increasing temperature. The low-frequency maximum is related to the tunneling of electrons between carbon coatings found on the surface of metal carbide nanoshells. The mid-frequency maximum is due to electron transitions between metal carbide nanoshells. The high-frequency maximum is associated with tunneling between the nano-grains and the carbon coatings.

This is indication of a step polarisation mechanism, whereby the concentration of dipoles, formed by tunneling, increases with increasing temperature. When solid dipoles are present in

materials (orientation polarisation), the static permittivity is described by the Debye formula and its value decreases with increasing temperature. In the frequency region above 1 MHz, an anomalous dispersion of the true permittivity was observed for all nanocomposite compositions – an increase in value with increasing frequency.

Although a detailed characterization, as well as mechanisms, are presented in the paper, there is no concrete application of the material in order to confirm the scientific contribution. Authors synthetized various materials, but without concrete application with scientific contribution.

I ask the authors to supplement the paper with the mentioned experiments, in order to make a scientific contribution of the paper.

I recommend the authors to review the manuscript and eliminate all shortcomings.

Accordingly, I recommend Editor to accept paper after minor revision (corrections to minor methodological errors and text editing).

Author Response

Please see in the attach.
